# Enhancing fieldworkers' performance management support in health research: an exploratory study on the views of field managers and fieldworkers from major research centres in Africa

Francis Kazungu Kombe  ,[1,2] Vicki Marsh,[1,3] Sassy Molyneux,[1,3] Dorcas Mwikali Kamuya,[1,4] Dorothy Ikamba,[1] Samson Muchina Kinyanjui[1,3]

¹KEMRI Wellcome Trust Research Programme, Kilifi, Kenya
²African Research Integrity Network, Kilifi, Kenya
³Centre for Tropical Medicine and Global Health, Nuffield Department of Medicine, Oxford University, Oxford, UK
⁴Ethox Centre, Nuffield Department of Population Health, Oxford University, Oxford, UK

**Correspondence to**
Francis Kazungu Kombe;
Kajoleh@gmail.com

## ABSTRACT

**Introduction** Fieldworkers are part of the system that promotes scientific and ethical standards in research, through data collection, consenting and supporting research, due to their insider cultural knowledge and fluency in local languages. The credibility and integrity of health research, therefore, rely on how fieldworkers adhere to institutional and research procedures and guidelines.

**Objectives** This study mapped out existing practices in training, support and performance management of fieldworkers in Africa, described fieldworkers' and their managers' experiences, and lessons learnt. A consultative process, involving field managers from 15 international health research institutions, was used to identify appropriate ways of addressing the challenges fieldworkers face.

**Methods** In phase 1, we conducted 32 telephone interviews with 20 field managers and 12 senior fieldworkers from 18 major research centres in Africa, Medical Research Council-UK and the INDEPTH Network Secretariat. In phase 2, we held a 2.5-day workshop involving 25 delegates, including 18 field managers from the institutions that were involved in phase 1 and 7 additional stakeholders from the KEMRI Wellcome Trust Research Programme (KWTRP). An earlier report from phase 1 was published in *BMC Medical Ethics* in 2015. Data transcribed from the interviews and workshop proceedings were analysed thematically using NVivo V.10 software.

**Results** Most institutions employed fieldworkers, usually with 12 years of formal education and residing within the geographical areas of research, to support studies. Although their roles were common, there were marked differences in the type of training, professional development schemes and fieldworkers support. Fieldworkers faced various challenges, with the potential to affect their ethical and scientific practices.

**Discussion** Fieldworkers undertake vital tasks that promote data quality and ethical practice in research. There is a need for research institutions to develop a structured support system, provide fieldworkers with interpersonal skills training, and provide space for discussion, reflection and experience sharing to help

## Strengths and limitations of this study

► The fact that participants of this study were drawn from West, South and East Africa makes the results of this study rich with an extended spectrum of experiences adduced, which provides deeper insights into the issues that were discussed.

► The use of consultative approaches, which allowed issues raised during telephone interviews to be followed up and discussed further in a workshop, provided a unique opportunity to crosscheck data from fieldworkers with their managers, and reflect on and unpack important issues, to identify relevant and applicable strategies for addressing challenges faced.

► Because all interviews were done by phone, important interview dynamics such as non-verbal cues may have been missed.

► Because our interviews focused on fieldworkers' challenges and potential opportunities, this could have raised fieldworkers' expectation to receive direct support to address the challenges they faced and could have contributed to over-reporting of the challenges they experienced.

► Because all interviewees were nominated by centre directors, who were the primary contact people, it is possible that interviewees could have reported what they believed would be acceptable to their directors due to power differentials between the participants (field managers and fieldworkers) and the centre directors.

fieldworkers tackle the practical and ethical challenges they face.

## BACKGROUND

High-quality data are a critical output of all sciences worldwide. Thus, research institutions put significant emphasis on generating data that are of highest scientific and ethical standards. In many research programmes in low-income and middle-income countries,



much of the data are collected by front-line staff, referred to as 'fieldworkers'. This cadre of staff is also often responsible for recruiting potential research participants, administering consent forms and following up participants during studies.[1] As such, beyond ensuring data integrity, fieldworkers play a key role in ensuring the ethical implementation of research protocols. The role of fieldworkers is particularly critical in international collaborative programmes with researchers drawn from outside local communities. In such settings, fieldworkers, who are normally from the local communities, can serve as the face of research programmes with the potential to significantly influence communities' trust in research.

In carrying out these complex roles, fieldworkers face practical and ethical dilemmas that require the balancing of institutional roles and guidelines against community expectations and fieldworkers' interests.[2] Hence, fieldworkers continuously make independent (often unconscious) decisions on how to apply 'ethical principles' on the ground. For example, fieldworkers reported challenges such as dealing with 'silent refusals'[3] (where participants avoid key study procedures but do not openly refuse) and with many requests for health and other social-economic support that are not included in the study protocol.[4–8]

### Capacity building and investment in research
Over the years there has been an increasing interest and investment in developing research capacity in Africa and other low-income and middle-income regions through long-term initiatives such as the WHO-Tropical Diseases Research Programme,[9] the European and Developing Countries Clinical Trials Partnership, and more recently the Wellcome Trust-Department for International Development (DFID)-funded DELTAS Africa ('Developing Excellence in Leadership, Training and Science' scheme). Efforts to strengthen research capacity are likely to continue as the trend towards greater cross-border data sharing, globalisation of clinical research and comparability of data quality at the global level continues.[10] Given that fieldworkers play such a vital role in health research, it would be expected that research programmes, in addition to training scientists, would also focus on building fieldworkers' capacity to carry out their duties in the most effective manner. However, fieldworkers' capacity and skillsets have received relatively little attention. The mechanisms and processes by which research institutions in Africa support fieldworkers in undertaking their duties remain unclear. Apart from the good clinical practice (GCP) guidelines, which focus primarily on clinical trials, no other guidelines are available that provide a framework for the content and approach for supporting the continuing professional development and career growth of fieldworkers. Understanding the challenges that fieldworkers face and the type of systems and level of support research institutions in Africa provide can enable us to identify opportunities for strengthening fieldworkers' scientific and ethical practices.

In this paper, we draw on our results from a two-phase qualitative exploratory study on the perceptions, experiences and views of field managers and fieldworkers on the structural and performance management support extended to health research fieldworkers, with the aim of informing institutional policies that promote research integrity and ethical practices for this critical cadre of staff.[11]

### Study objectives
This study mapped out existing practices in relation to training, support and performance management of fieldworkers in Africa, fieldworkers' and their managers' experiences, and lessons learnt. In addition, field managers from international health research institutions were involved in a consultative workshop to identify appropriate ways of addressing the challenges faced by fieldworkers working in research centres in Africa.

## METHODS
The study was planned over two phases and focused on understanding structural and performance management practices and their implications within institutions. In phase 1, which was implemented in February 2014, we conducted semistructured telephone interviews with 32 fieldworker managers and experienced fieldworkers across 18 health research institutions in 14 countries in Africa and the UK. In phase 2, which was implemented in July 2014, we held a 2.5-day workshop involving 25 delegates. Eighteen of these delegates were from the institutions that were involved in phase 1 (ie, one per institution). The workshop also included the director of training, three senior social scientists with long-term experience in managing fieldworkers (also co-Principal Investigators (PIs) of the project), the head of community engagement and two senior fieldworkers' supervisors from the KEMRI Wellcome Trust Research Programme (KWTRP).[11] Fieldworkers were not involved in the second phase.

This paper focuses on the results of phase 1 (telephone interviews). Issues that were identified during phase 1 were further discussed in more detail in the consultative workshop in order to build consensus and identify common recommendations and strategies for addressing the reported challenges. Thus, certain sections of the paper inevitably refer to views shared during the workshop. Results of the consultative workshop have already been published.[11]

### Patient and public involvement
Patients and the public were not involved in the design and conception of this study.

### Study site and selection of research institutions and interview participants
This study was led by a team based at KWTRP in Kenya. KWTRP is a long-standing international health research

institution with a wide range of existing scientific and capacity-building collaborations with other research institutes in Sub-Saharan Africa (see http://kemri-wellcome.org/). During the first phase, our sampling strategy for research institutions and relevant individuals within these focused on African institutions that manage health and demographic surveillance systems, given their employment of significant numbers of fieldworkers. We identified these institutions from our existing scientific and capacity-building linkages, including the INDEPTH Network (http://www.indepth-network.org), an international research network for health and demographic surveillance sites in Africa which encompasses all research centres in Sub-Saharan Africa that have health demographic surveillance systems.

From the original list of 34 institutions obtained from the KWTRP training department and INDEPTH Network site, we sent an email to the directors, with an information sheet about the study, to invite them to participate in the study. A total of 18 African international research institutions (see table 1) and the Medical Research Council (MRC)-UK and the INDEPTH Network Secretariat in Ghana responded positively. A total of 16 directors did not reply to our email and could not, therefore, determine why they did not respond; thus, the reasons for non-response are unknown. These were dropped after three reminders. The 18 African research institutions included a mix of institutions with varied years of operations that employed a diverse number of fieldworkers, engaged in a range of local and international collaborations, and specialised in different research programmes/thematic areas. This mixture of characteristics was important in maximising shared learning and experiences across institutions. In each institution, we wrote to the director and sought permission to interview one senior fieldworker and one field manager. In the letter, we particularly underscored the need to select a manager whose primary responsibility involved working directly with and managing fieldworkers, including offering support supervision, training and other forms of performance management.

## Data collection and analysis
### Phase 1: telephone interviews with managers and fieldworkers
Out of the 34 directors contacted, 18 sent names of two individuals (one senior fieldworker and one field manager) to be interviewed. We then contacted the proposed individuals and sought their consent to participate in a semistructured interview, using telephone or voice over internet protocol communication. Interviews were conducted using an interview guide (online supplementary appendix A) that was developed in consultation with the members of the study team and pretested before being used. In total, the 18 institutions that participated in the study employed a total workforce of 7839 staff, with 2505 (32.0%) being fieldworkers. Table 1 shows all the institutions that participated in the study, marked in dark blue (those that attended the workshop) and light blue (those that did not attend the workshop), with their names anonymised. In total, seven of the research institutions that participated in the interviews were from West Africa, four were from Southern Africa and seven were from East Africa. In addition, individuals from one regional research coordination institution from Africa and one UK research funding organisation were also interviewed. The institutions conducted research in various fields, including but not limited to infectious tropical and bacterial diseases (n=9), population health and HIV studies (n=7), social science and health systems research (n=9), clinical research/drug and vaccine trials (n=5), epidemiology prevention and control of malaria (n=9), maternal and newborn health (n=6), non-communicable diseases (n=3), fundamental sciences (n=1), human genetics (n=1), molecular biology (n=1), and clinical biology and bioinformatics (n=1). All the institutions received funding from external donors, mainly from the USA, UK and Canada. The number of staff employed per institution ranged from 45 to 1500 (mean 461), with the number of fieldworkers ranging from 30 to 630 (mean 147).

Each telephone interview lasted for approximately 1.5 hours. During the interviews, we sought to develop an understanding of current institutional practices and experiences in relation to training, support supervision and performance management of fieldworkers. Interviews with field managers explored issues related to processes of recruiting fieldworkers, the minimum qualification for fieldworkers, and the requirements for professional training as well as internal and external training given to fieldworkers. Additionally, issues around capacity building, support supervision and professional development for fieldworkers were discussed. Interviews with senior fieldworkers focused on their perceptions of the roles they play in research, challenges and mechanisms for addressing these challenges, career development opportunities, and other existing support systems. The first author conducted the interviews, which were voice-recorded, transcribed and immediately read through for verification and filling of any gaps by the interviewer before analysis.

### Phase 2: the workshop with field managers
A total of 25 managers from 15 of the 18 research centres that took part in the phase 1 telephone interviews attended the 2.5-day workshop to further discuss the phase 1 results. In total, the 15 institutions represented at the workshop employed a total of 6702 staff, with 2136 (31.9%) being fieldworkers. The institutions came from nine different African countries. To attend the workshop, invitations were sent directly to the individuals who participated in phase 1 interviews and requested to seek clearance from their line managers before attending the workshop.

The workshop agenda was developed collaboratively between the study team and an independent Kenyan

**Table 1** Main characteristics of institutions involved in the project

| Number | Name of institution | Year established | Main type of research | Staff (n) | Fieldworkers (n) | Fieldworkers (%) | Source of funding | Description of training given to fieldworkers | Description of career progressing system used |
|---|---|---|---|---|---|---|---|---|---|
| 1 | Research institution 1-WA | 1947 | Infectious tropical diseases. | 1104 | 301 | 27.3 | MRC-UK. | Core training is given to all fieldworkers with three levels, including basic, intermediary and advanced. Everybody who gets employed as a fieldworker must go through core training. | Successful completion of fieldworkers training is automatically awarded one salary grade promotion and allows fieldworkers to move through a career progression from A2 A3, B1 B2 B3, C1 C2 C3, D1 D2 D3, and E1 E2 E3. Some fieldworkers move up to C2 after the training because of the change in job description and responsibilities. |
| 2 | Research institution 2-SA | 1997 | Health and population, HIV studies. | 800 | 52 | 5.4 | WT, NIH and EGPAF. | A mass training is done for 5 weeks: 4 weeks course work and 1 week practical. Exam is done to assess skills gained and certificate issued on passing the exams. Those who pass are put on a waiting list and employed directly to studies when a vacancy arises. | A professional development process involving all staff is available. It is not specific to fieldworkers. |
| 3 | Research institution 3-WA | 1983 | Anthropology, demography, health economics, clinical trials. | 153 | 40 | 26.1 | NIH and USAID, WHO and IGAD. | A centrally managed training coordinated by a social scientist who takes through the fieldworkers on how to approach the community is usually given. | Fieldworkers are usually outsourced. No scheme of service is available. |
| 4 | Research institution 4-EA | 1970 | Infectious diseases, prevention and control. | 120 | 40 | 33.3 | NORAD, SIDA, EDCTP, WT, BMGF, WHO, TDR, GLOBVAC and Swiss TPH. | New fieldworkers are trained on the specific project and other training such as GCP, data management and ethics. | There is no career ladder, but if they (fieldworkers) improve their educational qualification they get an opportunity to apply for other higher positions. |
| 5 | Research institution 5-SA | 1996 | Malaria, HIV AIDS, vaccine trials, pneumonia, bacterial infection. | 200 | 100 | 50.0 | AECID, MOH and EDPTP. | All fieldworkers undergo project-related training, GCP, data management and ethics on being employed. | None. |
| 6 | Research institution 6-EA | 1989 | Immunology, TB, bacteraemia, HIV, social science, health economics. | 750 | 300 | 40 | WT, IAVI and NIH. | Fieldworkers training is centrally coordinated. All new fieldworkers undergo training in communication skills, research ethics, introduction to medical research, GCP and phlebotomy before being allowed to work in the field. | A career progression framework for fieldworkers exists, which stipulate the entry point and career progression subject to the attainment of job experience, inhouse training as well as external training, which could either be self-sponsored or programme-sponsored. |
| 7 | Research institution 7-EA | 1998 | Health system research, maternal and newborn health, and HIV. | 700 | 135 | 19.3 | DFID, INDEPTH, MOH, IDRC, DFID, RF, BMGF, FP7, USAID and CDC-RBM. | Fieldworkers who work in the DSS go through a refresher training every 6 months to share the challenges from the previous round and train them on new issues, discuss the challenges and how to address them. They (fieldworkers) are also reminded about ethical issues eg their ethical responsibilities, | An interviewer/field staff can start as a field interviewer (fieldworker) and then progress to the position of a field supervisor and sometimes even to the position of a field manager. HR has its arrangement but the structure is not formal. |

**Table 1** Continued

| Number | Name of institution | Year established | Main type of research | Staff (n) | Fieldworkers (n) | Fieldworkers (%) | Source of funding | Description of training given to fieldworkers | Description of career progressing system used |
|---|---|---|---|---|---|---|---|---|---|
| 8 | Research institution 8-EA | 1996 | Malaria, HIV, health and poverty, TB. | 800 | 183 | 22.9 | MOH, Swiss Institute, BMGF and EU. | Fieldworkers are given intensive training about how to work in the DSS and the procedures of doing DSS activities. They are also trained about ethics and their responsibilities. At the same time, other related issues concerning DSS activities are varied. | The training unit deals with all the training needs and capacity building. So a fieldworker can even be appointed to be a field supervisor or a field manager if they express interest. A structure is available. |
| 9 | Research institution 9-EA | 2011 | Health system research, reproductive health. | 250 | 100 | 75.6 | Wellcome Trust. | New fieldworkers are paired with more experienced fieldworkers for a given period of time. They are also required to attend mandatory meetings to share their experiences. After every DSS round, a 3-day refresher training is organised to recap on what they have done and solve any problems. | Training and capacity building are available for senior research staff but not for fieldworkers. |
| 10 | Research institution 10-WA | 2005 | Social scientist and clinical research. | 90 | 50 | 50.0 | MOH and LSHTM. | Training for fieldworkers is coordinated by the leader of the study team, so for every study there is a team responsible for coordinating the fieldworkers training, that train fieldworkers about field practices, community entry law and so on. | There is no formal structure for fieldworkers' career development. Only those with a degree and master's qualification can be promoted to senior positions. |
| 11 | Research institution 11-WA | 1994 | Malaria, TB, HIV, and maternal and child health. | 200 | 100 | 50.0 | LSHTM, Kwame Nkrumah University of Science and Technology, University of Ghana. | Fieldworkers receive training and retraining on the forms they use in the field, at the end of the round and the end of the year. Every week, they meet with their supervisors in the field to discuss any emerging issues. | Fieldworkers can be supported to undertake advanced training in nursing or laboratory, but there is no guarantee that their position would be reserved when they complete the course. A formal career progression structure is not available. |
| 12 | Research institution 12-EA | 1984 | Malaria, TB, HIV and emerging infection. | 1500 | 630 | 42.0 | CDC, EDCTP and other agencies. | Fieldworkers are trained on ethics, which entails GCP, then they are paired up with more experienced ones and then trained on relevant processes and specific study documents, tools, protocols, general orientation, confidentiality issues, and after this they are left to work alone. | There are promotions every year when staff is being evaluated and other conditions which staff should meet before the supervisor recommends them for promotion. No career progression framework for fieldworkers. |
| 13 | Research institution 13-EA | 1988 | Maternal and newborn health. | 380 | 105 | 27.6 | MRC-UK. | All fieldworkers undergo a GCP training before being trained on their specific study protocol. | Career progression depends on studies. Different studies have different requirement for career progression. Some of them may directly be organised by the programme, where a certain cadre of staff is selected for a specific training. There is no career progression structure for fieldworkers. |

Continued

**Table 1** Continued

| Number | Name of institution | Year established | Main type of research | Staff (n) | Fieldworkers (n) | Fieldworkers (%) | Source of funding | Description of training given to fieldworkers | Description of career progressing system used |
|---|---|---|---|---|---|---|---|---|---|
| 14 | Research institution 14-SA | 1995 | HIV and AIDS, malaria, non-communicable diseases. | 300 | 73 | 24.3 | WT and EU. | There is no standard structure. Once employed and taken through HR and study protocol induction, they (fieldworkers) are ready to go and work. There is no process to build fieldworkers' capacity to address practical and ethical challenges fieldworkers face. | No structure available, but if there is a new position, a fieldworker who has the appropriate qualification can apply. There is no formal approach to supporting fieldworkers. Sponsorship is competitive against all staff members. |
| 15 | Research institution 15-SA | 1979 | TB, HIV, NCD, clinical and epidemiological research. | 184 | 73 | 39.7 | WT, LSHTM, BMGF and EU. | Fieldworkers get trained on the demographic surveillance system, its basic outline and principles because most studies are linked to the continuous registration system. They (fieldworkers) are also trained on HIV counselling and testing and data entry. | No formal career progression pathway exists. |
| 16 | Research institution 16-WA | 1992 | Vaccine and drug trials, fundamental sciences, mosquitoes genetics, parasites genetics, drug resistance (PCR, in vitro tests and so on), malaria immunology, malaria and pregnancy trial. | 45 | 30 | 66.7 | NIH, MOH and EDCTP. | Fieldworkers are given protocol-specific training and told about their responsibilities and the objectives of the study. | Training/career development programme is in place to increase the number of qualified scientific and technical personnel capable of conducting independent medical research. This includes postdoctoral training—PhD level: parasitology, cellular biology, pharmacology, immunology; master's of sciences: clinical research, clinical pharmacology, public health, parasitology, entomology and nutrition. No specific framework for fieldworkers. |
| 17 | Research institution 17-WA | 1998 | Maternal and child health, tropical diseases; malaria, reproductive health, communicable diseases; non-communicable diseases; human genetics; health systems research, social and behavioural studies, poverty and equity studies. | 263 | 193 | 73 | Ghana Health Service, Rockefeller Foundation, Population Council- USA, USAID. | Fieldworkers training depends on the nature of the project, so for instance, if they are doing a survey on the health system, they will be taken through that and the overview of the overall centre, community engagement and ethics. The project staff and the PI undertake the training. | If a junior/front-line staff goes for further studies and has worked for 10 or more years and the field of study is relevant to their line of work, then they may be supported by the centre. There is no formal structure to support fieldworkers. |

**Table 1** Continued

| Number | Name of institution | Year established | Main type of research | Staff (n) | Fieldworkers (n) | Fieldworkers (%) | Source of funding | Description of training given to fieldworkers | Description of career progressing system used |
|---|---|---|---|---|---|---|---|---|---|
| 18 | Research institution 18-WA | 2007 | Molecular biology, immunology, clinical biology and bioinformatics. | Staff are contracted from other organisations. | Fieldworkers are contracted through other organisations on a need basis then laid off at the end of a study. | Depends on the project. | National government, UNIAIDS, Liverpool University, NIH and Fogarty Foundation. | New fieldworkers are usually given a standard operational procedure then shown what they are going to do. After that, they can be allowed to work in the field. | Fieldworkers are contracted through other organisations on a need basis, then laid off at the end of a study. |

A discussion with MRC-UK and the INDEPTH Secretariat in Ghana was also held to crosscheck the information gathered from the interviews held.
Total: 7 research institutions from West Africa (WA), 4 research institutions from Southern Africa (SA) and 7 research institutions from East Africa (EA), and 1 regional research coordination institution from Africa and 1 UK research funding institution.
Dark blue refers to those that attended the workshop and light blue refers to those that did not attend the workshop.
AECID, Spanish Agency for International Development; BMGF, Bill and Melinda Gates Foundation; CDC, Center for Disease Control; CDC-RBM, Center for Disease Control- Roll Back Malarai Partnership; DfID, Department for International Development (UK); DSS, demographic surveillance system; EDCTP, European and Developing Countries Clinical Trials Partnership; EGPAF, Elizabeth Glaser Pediatric AIDS Foundation; EU, European Union; FP7, European Union Research and Innovation Funding Programme for 2007-2013; GCP, good clinical practice; GLOBVAC, Programme for Global Health and Vaccination Research; HR, Human Resources; IAVI, International AIDS Vaccine Initiative; IGAD, The Intergovernmental Authority on Development; LSHTM, London School of Hygiene and Tropical Medicine; MOH, Ministry of Health; MRC, Medical Research Council; NCD, Non Communicable Diseases; NIH, National Institutes of Health; NORAD, Norwegian Organization for Development Cooperation; PI, Principal Investigator; RF, Rockefeller Foundation; SIDA, Swedish International Development Cooperation Agency; Swiss TPH, Swiss Tropical and Health Institute; TB, tuberculosis; TDR, Tropical Diseases Research Program; UNAIDS, The Joint United Nations Programme on HIV and AIDS; USAID, US Agency for International Development; WT, Wellcome Trust, UK.

consultant, who acted as the main workshop facilitator and who had been selected based on his indepth experience of participatory methods and prior familiarity with research institutions in Africa. Overall, the workshop combined information sharing and large group discussions on day 1 with a series of participatory consultative and consensus-building activities over the following days. On day 1, two sets of presentations were made and discussed in plenary. The first presentation described the phase 1 results, with discussions on the range, strengths and challenges of fieldworkers' employment and support structures across institutions. The second presentation highlighted the results from earlier social science research at KWTRP and other research sites in Africa on the complex nature of fieldwork, including practical, social and ethical challenges that fieldworkers often face as part of their routine work, particularly within low-income communities and households. Discussions that followed the presentations focused on the relevance and importance of these challenges across sites.

### Data analysis

Data analysis in this study followed a framework analysis approach.[12] Transcripts were independently read and discussed in depth by two members of the study team (FKK and VM) to develop an initial coding framework, drawing on the main questions raised during interviews (see online supplementary appendix A) and themes that emerged during discussions. FKK used NVivo V.10 software to organise data across the transcripts under these codes, adding codes for new issues emerging from the data. Data analysis charts were developed by FKK and VM to account for the codes under themes by the participants and support interpretation of patterns within the responses, drawing on the social science and research ethics literature.

Preliminary analysis of the telephone survey data was used to inform about the type and range of fieldworkers' support supervision and training practices as well as performance management strategies. The themes generated from this initial analysis were explored further and discussed in detail during the consultative workshop, with a view for consensus building and checking for similarities and variations across different research institutions. Emerging new themes were also discussed and explored using a combination of qualitative participatory approaches, including large and small group discussions, round table and moderated discussions.

Moderators were careful to discuss ideas in detail without influencing the direction and outcome of the discussion. During the analysis, we compared and contrasted the fieldworkers' and field managers' views based on specific themes. Through this, we realised there were issues that fieldworkers and field managers shared the same views and others where they had different views. In this paper, we have tried to present these similarities and contrasts together, rather than separately. We believe this will be much easier for our readers to follow than it

would have been if we separated the views of fieldworkers from those of the field managers. Finally, although our participants are classified as fieldworkers and field managers, it is important to note that questions in the interview guides that were used to interview them were very similar. Despite the similarity of the interview guide used, interviews with the field managers were considerably longer, allowing for a more indepth exploration of issues. This makes it logical to analyse and present these data together, as shown in the next section.

## RESULTS

In this section, we describe and discuss the results from the telephone survey, and where applicable the highlights of the workshop, across three inter-related themes that were explored or emerged as important across this study. These themes are (1) the current institutional structures and systems that support fieldworkers across the institutions involved in the study; (2) participants' perceptions of work-related challenges for fieldworkers and their relationship to support structures; and (3) views on the ways in which challenges for fieldworkers might be addressed, including agreement on institutional and wider recommendations.

### Current institutional structures and systems for fieldworkers

Across interviews and the subsequent workshop, it became clear that there is a large amount of diversity in the names, tasks and institutional structures that govern and support the way that fieldworkers operate in the institutions involved. For example, during interviews, senior fieldworkers described the existence of many different tasks and titles, including fieldworker, field enumerator, interviewer, field officer, tracer, community technician and field assistant, which were often used interchangeably, within and across different centres. Similarly, the way that fieldworkers were recruited, contracted, trained and supported in their work was highly variable, between projects within an institution and between institutions. Table 1 describes these parameters across the research institutions involved.

As table 1 shows, some form of inhouse fieldworkers training was common across the institutions, with training mainly focused on the skills needed to undertake study-specific tasks, such as using particular data collection tools. In some cases, training in GCP was also offered, particularly to support fieldworkers undertaking informed consent processes. Often, such training was generally the responsibility of individual study teams within institutions. Four out of the 15 institutions (27%) confirmed having a centrally coordinated training approach, 3 of which covered a more comprehensive set of topics including data collection skills, communication skills, informed consent processes, GCP and research ethics more generally. The other sites described conducting some kind of ad hoc training based on a specific study requirement or had no training plans for fieldworkers.

One area of fieldworker performance management which was rarely discussed and generally under-recognised across institutions was the concept of fieldworkers' professional career progression or continuing professional development. While such measures were largely routine for other cadres of research staff within institutions, they had not been applied to fieldworkers. Many felt this would be an important step towards recognising the important role fieldworkers play and in motivating them to maintain a high professional standard of work, but challenges were seen in sustaining such an initiative given the typically short-term contracts offered to fieldworkers, based on similarly short research funding cycles.

> It is good to establish a professional development scheme for fieldworkers. However, such a system may not be sustainable, given the dynamic employment terms of fieldworkers and researcher in general…So although it is a good concept, it may be difficult to implement it [the scheme] within the context of research that relies on funding that is not always guaranteed. (FWM, SA)

> The first three letters of the initials in the quotes stand for the position of the interviewee, that is, SFW for senior field worker and FWM for field worker manager. The last two letters represent the country, for example, KE for Kenya, SA for South Africa and GN for Ghana.

### Participants' perceptions of work-related challenges for fieldworkers and their relationship to support structures

Across the following paragraphs, we describe the forms of challenges that fieldworkers and those working closely with them described during interviews, and that were taken forward for discussion at the workshop. In describing these, we highlight the types of institutional support that influence these challenges, including structural influences from human resource management strategies and research funding cycles, and more ad hoc often individual efforts to manage complex challenges on the ground.

#### Using different 'labels' for fieldworkers

Discussions during interviews and at the workshop drew out differences in opinions on the importance of the 'labels' that were used to describe the work we describe here as 'fieldwork'. At a most pragmatic level, some fieldworkers felt that the most important issue was having a job, rather than being concerned about the title that was associated with that job. Practically, this attitude seems to hint at potentially problematic power imbalances for fieldworkers in relation to the institutions in which they were employed. Other fieldworkers felt that using different 'labels', particularly within one institution, could generate confusion about roles and lead to employment conditions being less transparent:

> It [roles taken by field workers] is not different. These are the organisation's terms…community technician,

field worker, follow up staff, field technician…I mean it is just the same thing…it's just their [the organisation] way of making us look different, but we are the same. (SFW, KE)

Many field managers recognised the phenomenon of multiple labels as being used for slightly different forms of 'fieldwork'. There was, however, an initial divergence of opinion about the value of harmonising job titles. Some felt that this variation underlined important, even if small, differences in fieldworkers' current roles and prior experience or qualifications, which might not be obvious to all team members. Others felt there was little need for multiple titles, given the potential for confusion and conflict this might present. As will be described in the later section on recommendations, developing clear and strategic approaches for fieldworkers' roles within an institution emerged as an important feature of good practice for fieldworkers' employment. For many, harmonising these structures and processes across institutions was also seen as valuable. Field managers felt harmonising fieldworker employment terms, training and skills would promote cross-study and institution transfer of fieldworkers and address the challenge associated with a high turnover of fieldworkers due to short-term contracts.

### Seeing fieldwork as a short-term activity

A core tension that was strongly expressed in interviews and the workshop was a general lack of recognition for the work that fieldworkers undertake within research institutions. Fieldworkers themselves saw their work as important, and as fundamentally contributing to answering important research questions, and promoting the reputation of institutions, careers of scientists and other cadres of research staff. There was, however, some frustration and perceived unfairness around the lack of opportunities for fieldworkers to progress within their area of work, or achieve recognition over sometimes periods of many years working in this role. Experiences were shared in which scientists who joined research centres were able to move ahead to higher positions through focused capacity-building strategies, while fieldworkers' positions remained static. Some fieldworkers described working for an institution for more than 10 years without more than basic training or promotion. Perhaps not surprising, given the way that 'fieldwork' has traditionally been viewed within research institutions (ie, as short-term and relatively non-technical work), the institutions involved in this study had not developed strategic approaches for fieldworkers' career development.

In response to these frustrations, some fieldworkers had taken up self-funded part-time training courses outside their existing jobs in the hope that this would allow them to progress within the institution. This signified the importance of how fieldworkers valued education and training as an important element of their career advancement. In practice, however, institutions did not have structures that would allow fieldworkers to move into roles in which such

new skills could be used. Against this background, most fieldworkers reported feeling demotivated, including a feeling of low self-esteem and disillusionment:

Once you are employed as a fieldworker, you will always be a fieldworker [regardless of your qualifications]. (SFW, GN)

While some field managers supported this view, some argued that career development was a matter of personal motivation and determination. They explained how they (managers) had moved through the career ladder to become field managers, often starting their careers as junior fieldworkers and gaining extensive fieldwork experience along with their career progression to their current positions, with the same limited training and support. Most managers had therefore been part of the system for a long time and were able to share their views both from the perspective of fieldworkers as well as field managers, hence extending the spectrum of experiences adduced and providing deeper insights into the issues that were raised. Given these challenges, which workshop participants saw as fundamentally unfair and as potentially importantly undermining the quality of research at its roots, participants agreed that fieldworkers should have greater recognition for the role they play within research institutions. They also agreed that it was important for institutions to support some form of career progression, for example, within a specialised category of staff supporting fieldwork. Providing fieldworkers with generic knowledge and skills on core research areas as part of such a scheme would enable them to transfer between studies at the end of a funding cycle. This would enable research centres to retain experienced fieldworkers, a policy that was thought likely to promote high scientific and ethical standards, and give fieldworkers more stability, skills and recognition.

### Responding to complex challenges on the ground

In both the telephone interviews and the subsequent workshop, fieldworkers and managers described facing often very difficult challenges that seem to be implicit in the nature of much fieldwork. Some of these were practical, including the physical burden of fieldwork, such as the experience of long working days with long distances to cover on foot, risks of injury through dog or snake bites, and physical assault in less secure settings. Many also described emotional burden including anxiety, fear, sadness and sometimes helplessness when interacting with individuals and families facing severe social and economic hardship. Within the community, fieldworkers said they were perceived as 'fellow community members' and, at the same time, the 'face' of the institutions that employ them, generating expectations of support that were beyond their mandates from the community members they visited.[13]

These challenges have been well described in the literature on global health ethics, including from social science research at KWTRP, and our results in this

study underline the existence of these important challenges.[1 4 5 13 14] The fact that our study draws on experiences from 18 major research centres in Africa and found similar challenges is an indication of how common and important those challenges are. Within this literature, there is increasing recognition of the moral work that fieldworkers are involved in during their everyday work[1 3 15 16] which can directly impact the science and ethics of the studies they work for.[5 17] For example, a key issue in the literature concerns the negotiation of informed consent processes, where fieldworkers balance perceived pressures to recruit participants according to targets set by managers and the extended period that may be needed to explain complex and often unfamiliar topics within consent forms to household members. Besides, community members may feel that by agreeing to participate in studies, they will develop some leverage to access greater institutional support, including through their relationships with fieldworkers.[4 5 18–20] Similar ethical challenges experienced by fieldworkers have been reported in high-income settings, including the USA, with a specific focus on how fieldworkers' and study coordinators' practices can affect the protection of research participants.[21] Although it could be argued that these settings are contextually different, it would be interesting to examine why fieldworkers in high-income countries experience the same ethical challenges as those in Africa, despite them being better trained, educated and resourced than their counterparts.[2 21]

A particularly difficult situation that fieldworkers involved in routine demographic surveillance system activities in this study described was the need to directly interact with bereaved families and seek information about deceased relatives as part of conducting verbal autopsies (interviews held with close family members of a recently deceased person to try to establish the cause of death).[22 23] Such emotionally challenging interactions generated distress and worry for fieldworkers and could chronically undermine performance, leading to 'burnout':

> Yeah but you see what, after maybe going through these challenges sometimes they affect you or they affect your output. (SFW, SA)

> …I felt bad, and you know the following day I was to go back to work, and there is no supportive counselling or something like that… So as fieldworkers sometimes we just burn out and sometimes with this kind of trauma, it becomes difficult…we go through a lot. (SFW, KE)

During workshop discussions about support mechanisms to help fieldworkers address these emotional challenges, field managers were not able to identify existing strategies, as was also reported by fieldworkers during the telephone interviews. Some managers described generally relying on fieldworkers' judgements in handling these difficult situations.

> It is difficult to anticipate what challenges fieldworkers are likely to face…and even if they were to share them [the challenges] some of them are too complex to resolve. It is, therefore, easier to just let them [fieldworkers] solve them spontaneously and trust that they use enough wisdom to do the right thing. (FWM, KE)

Experiences shared by fieldworkers revealed that the issues were usually quite complex and intertwined with cultural, structural and emotional sensitivities associated with the need to balance community and cultural expectation against the policies and guidelines of research institutions. Thus, fieldworkers were often expected to strike a delicate balance, sometimes without the knowledge of how acceptable their actions were. Despite these actions having the potential to directly affect the scientific and ethical standards of data that fieldworkers collect, structural support and referral systems were reportedly not available. This was well captured by one senior fieldworker who said:

> …sometimes, you get that huruma [pity], you go into your pocket and give out something…if you have a human heart, sometimes some situations are touching. (SFW, KE)

During interviews and workshop discussions, there were no accounts of existing internal or external institutional mechanisms to support fieldworkers to specifically deal with these types of emotional challenges. Although effective support supervision and sharing these challenges during team meeting were identified as a potential avenue to identify strategies to deal with these challenges, the approaches used to supervise fieldworkers were noted to be significantly different across institutions and not centrally coordinated within institutions, making it difficult to reflect on their effectiveness. Workshop participants therefore generally felt there was a gap in fieldworkers' support structures that should be addressed. These challenges were seen as occurring across all the institutions involved in the study.

### Ways of strengthening institutional support

Based on the outputs of discussions described so far, two core institutional fieldworker support strategies were raised, discussed and agreed on over the course of the workshop. The first concerned adapting the *content of skill-building initiatives* to support fieldworkers to manage their interactions with community members and household members in ways that could support good scientific and ethical practice, seen from multiple perspectives. The second strategic support mechanism addressed the wider question of how professional support within institutions could support fieldworkers more *structurally* towards these aims, including aspects of performance management that commonly exist for other groups of research staff. Workshop participants agreed that 'performance management' in this respect would include aspects of

recruitment, training, supervision, support, assessment, and career or professional development. Specifically, delegates emphasised the need to strengthen fieldworkers' interpersonal skills training to ensure they are well equipped to deal with cross-cutting issues around communication, community engagement, and building trust and respect, which are fundamental in consenting and quality data collection. Fieldworkers themselves believed that more support would improve their skills, and accordingly the quality of the data they collect. The need to develop a harmonised training curriculum with core training modules across Africa was identified.

In this study, we noted a mismatch related to lack of appropriate training for fieldworkers: fieldworkers' limited level of education and the complexity of the tasks they are expected to undertake, including but not limited to translating study information to study participants, and understanding institutional and international ethical guidelines and applying these when interacting with study participants. These results support the need for research institutions in Africa to invest in training and building the capacity of fieldworkers, to enhance the standards of the work fieldworkers undertake and provide optimum support for research.

### Agreement on recommendations

Given the issues discussed in the preceding paragraphs around institutional fieldworkers' support structures, participants at the workshop agreed there was an urgent need to introduce more standardised and comprehensive approaches, focused on important areas that affect fieldworkers' performance and practices, including career progression, training/capacity building and management of emotional and ethical challenges. It was felt that strategies could build on the positive experiences of research institutions that had already begun to institutionalise such approaches. Participants recommended a pragmatic approach, aimed at empowering and building the knowledge and skills of fieldworkers, including processes to support continuous learning through the regular reflection of the practical and ethical issues they face in everyday experience. In this way, an important element of training and support supervision was described as enhancing skills to recognise and respond to these ethical issues. The specific recommendations made are described and discussed in the following paragraphs.

### Creating space for experience sharing and problem solving

One challenge identified during the survey was lack of psychological support after experiencing difficult and sometimes traumatic experiences, such as some interviewees breaking down during verbal autopsy interviews, losing a study participant when close relationships had been built and even being physically attacked by emotional/aggrieved participants during interviews. Long working days with long distances to cover on foot, risks of injury through dog or snake bites, and physical assault in less secure settings were also said to be emotionally

overburdening to fieldworkers. Documented evidence shows that stressful and traumatic experiences can alter the way people view themselves and the world around them, and alter how they process information and the way they behave and respond to the environment around them.[24] A strategy described in the literature to address this challenge is the use of regular debrief sessions, where fieldworkers are given opportunities to share experiences and consider the best approach to address challenges. During the workshop, participants recommended that field managers should create such opportunities for fieldworkers to talk about distressing or worrying issues encountered at work, through forms of feedback and reflection. This approach also requires a support supervision mechanism that is sensitive to the complex nature of the experiences fieldworkers face. This supports True *et al's*[25] arguments that having access to colleagues to discuss issues and having effective support supervision acted as a buffering factor for research front-line staff who did not engage in ethical misconduct, despite facing the same pressures to meet recruitment goals and other stressors of conducting community-based research. Combining different ways of knowing and learning will enable fieldworkers to standardise the way they handle practical and ethical issues they encounter, even where these present as dilemmas with limited information on the best course of action. At the same time, in some cases, more traditional approaches to managing emotional distress, such as staff counselling, might also be needed.

### Increasing attention on soft skills for fieldworkers

As reported by both fieldworkers and field managers, advancement has been made in the area of technical training for research staff, including fieldworkers. GCP, health and safety, good clinical laboratory practice (GCLP) training has become a standard approach for clinical study site preparation. While this training enables studies to adhere to international GCP/GCLP standards, they rarely equip fieldworkers with the necessary skills to enable them to deal with practical local challenges they face on the ground. There is, therefore, a need for fieldworkers to undergo more generic interpersonal skills training, such as communication skills, basic psychology and counselling, and community engagement techniques to enable them to appreciate the practical local situations and develop skills to deal with the ground challenges.

### Having career development guidelines that are responsive to fieldworkers

In a previous commentary by Kombe *et al*,[11] which reported the views of field managers who attended a consultative workshop as part of this study, important recommendations made by the managers were presented, including the need to (1) increase institutional recognition of fieldworkers' roles and the need for systematic and comprehensive capacity building to ensure adequate resources are allocated and that capacity-building activities are well described in research proposals and grant applications;

(2) develop common areas of a core curriculum for fieldworkers' capacity building to enhance quality of training processes and build their knowledge in basic biology, research approaches and methods, research ethics and research regulatory frameworks, as well as data collection and documentation, respectful communication, and being aware of and managing ethical challenges and issues in practice; and (3) increase emphasis on fieldworkers' career development, including developing regionally accredited training to support fieldworkers' professional development to increase individual motivation and enhance capacity for employment across different research organisations.[11] These recommendations should be considered in order to address the challenges presented in this paper and enhance the overall quality and integrity of research in Africa. Similar strategies have been proposed by True *et al*[2] to address the challenges faced by community members employed as research staff in the USA.

## DISCUSSION

The important role fieldworkers play in health research can be enhanced if they are given adequate support. The challenges they face require a combination of knowledge, skills, attitude change and ethical values that can be enhanced through structured institutional support and instilling a sense of professional responsibility and moral values to fieldworkers, through ongoing support supervision, interpersonal skills training, and creating space for consultative discussions and experience sharing.

In this study, fieldworkers were reported to provide important support for research. From the institutions that participated in the study, fieldworkers were employed to undertake diverse tasks and were given a range of different titles. All participants that were interviewed acknowledged the fact that the tasks undertaken by fieldworkers were critical in supporting health research. On the other hand, fieldworkers reported experiencing a myriad of challenges, a fact that was supported by the fieldworker managers interviewed, and strongly deliberated on during the workshop, during the second phase of this study.

Despite the above-perceived importance of the tasks fieldworkers undertake in supporting health research, there were no centrally and institutionally coordinated systems or structures that were reported to have been put in place to provide the much-needed support for fieldworkers in most of the institutions involved in the study. Fieldworkers' coordination and support were largely left in the hands of study-specific field managers and supervisors, who often focused on the outputs of the specific studies they coordinated, with much less focus on generic support and supervision systems. Institutional systems to support fieldworkers to undertake their tasks effectively were reported to be limited or often unavailable. Furthermore, in spite of the myriad of challenges fieldworkers reported to experience in their day-to-day activities,

which were confirmed by their field managers during the telephone interviews and the consultative workshop, no institutional systems were reported to be in place to support fieldworkers in addressing such challenges. Fieldworkers were, therefore, essentially expected to use their discretion, cultural competence and moral reasoning to navigate through the many practical and ethical challenges they faced. Given the limited support, education and ethics knowledge and competence fieldworkers possessed, this situation points to serious implications on the quality of the data fieldworkers collect, their ethical practice and the overall integrity of the health research enterprise fieldworkers are expected to support.

This study interviewed participants via telephone. Although this could not have necessarily affected the quality of the data collected, important qualitative interview dynamics, which can provide important insights into the data analysis process, could not be identified and considered. However, the triangulation of different study approaches, including the consultative workshop with field managers, provided an important approach for validating the data collected and strengthening the study design.

The results from this study underscore the need to establish and strengthen institutional support for health research fieldworkers. Although a lot of studies have been done to understand the moral and ethical challenges and dilemmas faced by fieldworkers, very little has been done to explore which fieldworkers' institutional support systems exist and what implications such systems have on fieldworkers' scientific and ethical practices. There is, therefore, a need for further research to develop a deeper understanding of this area. This study has highlighted important gaps related to the management and coordination of fieldworkers that require further assessment and strategic consideration. Furthermore, the strategic solutions and recommendations identified in this study call for the need for health research institutions in Africa and funders of health research to strongly consider developing institutional systems and structures to facilitate centralised coordination and management of fieldworkers by health research institutions in Africa. As stated by Alexander and Richman,[26] strengthening the capacity of fieldworkers has the potential to increase the protection of research participants, enhance research integrity and promote the collection of valid data. Efforts to empower fieldworkers to make them better able to deal with these challenges should therefore continue.

This was an exploratory study, hence the results are not generalisable. However, they do offer important insights into fieldworkers' and field managers' views and experiences of fieldworkers' performance managing practices for institutions that come from similar settings as those that participated in this study. Although some of the proposed approaches such as interpersonal skills training, centralised coordination of fieldworkers training, fieldworkers' scheme of service and regular discussions to address fieldwork challenges have been

ongoing in institutions such as KWTRP, in Kenya, lack of documented evidence makes it difficult to say how effective and sustainable such approaches are. More research, therefore, needs to be done to explore the feasibility of the proposed strategies and generalisability of these study results within a wider context in Africa. Finally, the proposed solutions may not solve all the challenges fieldworkers face, but they are certainly an important step towards improving the scientific and ethical standards of health research output from the African continent. The value of this study is not that every research centre in Africa can identify with every issue raised here, as this would be difficult due to the variety of research centres in the continent; rather, we hope that by raising the issues highlighted here, even those centres that were not involved in the study will be provoked to examine their fieldworkers' support system and see if any issues raised here apply to their setting. Furthermore, empirical studies should be done to identify which approaches are most effective in enabling research fieldworkers to deal with the myriad of challenges they face. Such research will go a long way in improving the scientific and ethical standards research, not only in Africa but internationally, given the critical global role of Africa in research.

**Acknowledgements** We acknowledge and greatly appreciate the support of all research institutions' directors who allowed their field managers and fieldworkers to take part in this project, and all the field managers and fieldworkers who participated. We are extremely indebted to Professor Mike Parker, Director of the Wellcome Centre for Ethics and Humanities and Director of Ethox Centre at the University of Oxford, for his support and guidance during the project; and all the senior field managers and delegates who attended the consultative workshop. The manuscript is published with the support of the director, KEMRI.

**Contributors** All authors were involved in conceptualising the idea and contributed to writing the manuscript. FKK, a male senior community facilitator with a wealth of experience in health research ethics, community engagement and communication skills, who worked as a fieldworker himself, before taking a managerial role at KWTRP, contributed to the planning, conduct and reporting of the work described in the article and is responsible for the overall content. VM, a female senior social scientist at KWTRP and Oxford University, and DI, a female career development officer at KWTRP, and career counsellor, assisted with reviewing the transcripts and the development of themes and subthemes during the analysis. SM, a female senior social scientist at KWTRP and Oxford University, DMK, a female social scientist at KWTRP and Wellcome Trust Society & Ethics Fellow, and SMK, a male immunologist and head of training at KWTRP, contributed to designing the initial research ideas, reviewed the data collected and were involved in drafting the initial manuscript and edited subsequent versions. All authors reviewed and approved the final manuscript for publication. FKK accepts full responsibility for the work and the conduct of the study. He had full access to the data and controlled the decision to publish. FKK attests that all listed authors meet the authorship criteria and that no others meeting the criteria have been omitted.

**Funding** This project was funded by Global Health Bioethics Network-UK (Wellcome Trust grant no OXF-GHB02) and supplemented by KWTRP Capacity Building Strategic Award (Wellcome Trust grant no 084538). The research and capacity-building activities of the Global Health Bioethics Network are supported by a Wellcome Trust Strategic Award (096527).

**Competing interests** None declared.

**Patient consent for publication** Not required.

**Ethics approval** The paper is published with the permission of the centre director, KEMRI. All the participants who participated in the telephone survey as well as those who attended the consultative workshop gave individual consent to participate in the study and the workshop and permitted the publication of this manuscript. Permission to conduct this study was granted by a review conducted by the Centre for Geographic Medicine Research, Coast (CGMRC) - Centre Specific Committee, and final ethical approval was given by the Research and Scientific Review Unit (SERU) at the Kenya Medical Research Institute (KEMRI).

**Provenance and peer review** Not commissioned; externally peer reviewed.

**Data availability statement** Data are available upon reasonable request.

**ORCID iD**
Francis Kazungu Kombe http://orcid.org/0000-0003-0390-2336

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
