## [Reviewer comments · BMJ Open]

ARTICLE DETAILS

TITLE (PROVISIONAL)	Enhancing Fieldworkers' Performance Management Support in Health Research: An exploratory study on the views of Field managers and Fieldworkers from major research centres in Africa
AUTHORS	Kombe, Francis; Marsh, V; Molyneux, Sassy; Kamuya, Dorcas; Ikamba, Dorothy; Kinyanjui, Samson

VERSION 1 - REVIEW

REVIEWER	Hilde Buiting Netherlands Cancer Institute, Netherlands
REVIEW RETURNED	21-Dec-2018

GENERAL COMMENTS	 - In the research question I suggest to incorporate more clearly that this is about the experiences of fieldworkers/workshopmembers - If possible, I would add something in the discussion in why education is particularly important in sub-saharan countries - This is a qualitative study, so outcomes remain subjective, which is OK - I could not find any limitations of the study/reporting checklist - I could not find a clear data analysis section Enhancing fieldworkers' performance management support in health research: Views of field managers and fieldworkers from major research centres in Africa. General: In this study, the practice of training, support, and performance management of fieldworkers in sub-Saharan Africa was described, using in-depth interviews and a workshop. Paying attention to the role of fieldworkers is worthwhile, given the important role they often have in research-projects. The study showed that structured support, supervision and soft skills would be worthwhile to develop further. The study objective is relevant given the impact the work of fieldworkers can have on the data quality. The study is concrete, and well-written. I however, also have several questions for clarification and improvement. Abstract: - In the Introduction the authors write that 'fieldworkers play a critical role'...; I would rewrite this a bit, in 'are part of the system that promote scientific and ethical standards' (more than a role). - In either the study objective or introduction, it would help the reader if the authors would give a short definition of a
---

fieldworker. I think it is more or less similar as research assistants in European/US projects?

- In the results-section again, I feel that 'role' is maybe not the right term in this context. Perhaps, their tasks (such as...) were common is easier to understand.

- I wondered whether the options in an African setting are completely different compared to a US/European setting, and whether this is specific reason to pay more attention to the fieldworker tasks and education than in European countries, as most people often 'promote' as a researcher. Or whether the problem is especially related to the African/financial context. It would be interesting if the authors could reflect a bit more in the discussion section about this aspect also.

- In the results/conclusion I wondered whether some of the fieldworkers also indicated to prefer more education on 'content-specific' topics.

Background:

- Very clearly written; the relevance of doing such a research-project has been made clear as well.

Methods:

- General: While reading this method-section I wondered how fieldworkers were selected; what are the inclusion criteria of being a 'fieldworker', are there large differences in tasks across fieldworkers? What kind of research projects I can think of. What type of activities those fieldworkers generally do? Although I understand that the main focus of this paper is on training, I'm very interested about what kind of people we are talking about.

- Page 6, line 1-9; I wondered what type of research centres were included (could be described in text more clearly)?

- Can the authors describe the data analysis more concrete?

Findings and discussion

- General: I would suggest to split findings and discussion in two separate sections, as this will probably make the interpretation and discussion more easily to follow. I also became a bit confused because of all different ways of numbering.

- Can the authors explain why they decided to present the findings of interviews and the workshop together as well? I can imagine, that it might be good also to present the data of the workshop later-on, as this may partly result from the interviews, and this is a complete different type of data collection. Again, splitting those sections may help in interpreting the data more easily.

- Page 13, line 16 and further. Could the authors give some explanations of the type of difficult/traumatic life experiences? This will facilitate reading.

- Page 13, line 48, I can imagine that it very much depends on the type of fieldwork as to whether you need those courses such as GCLP. In this section as well, I'm really thinking of what type of work those fieldworkers need to do.

- Page 13, line 48; can the authors give some local challenges? I can imagine that this for instance also has to do with culture, access to material, et cetera.

Conclusions

- I would first start with the notion that this is all about the experiences/opinions of fieldworkers / workshop-members. E.g.

	state that fieldworkers themselves believe that more support would improve their skills, and accordingly the quality of the data collection. In the second sentence many aspects are mentioned. I would suggest to focus on a few aspects and then elaborate on this a bit more. I further wondered whether the recommendations are feasible in this setting. Could the authors reflect a bit more on this, also?
--	---

REVIEWER	leslie alexander Bryn Mawr College Graduate School of Social Work & Social Research USA
REVIEW RETURNED	21-Mar-2019

GENERAL COMMENTS	1)clarify Ns throughout 2)justification for merging fieldworker data with field manager and workshop data - need to be separated in findings - more excerpts from both managers and workers in findings 3) How this study extends findings of 2015 report in MMJEthics in 2015? This is an interesting article, from a group of researchers who have done a lot of publishing on fieldworkers in Africa. I do have some concerns that lessen the impact that the study has. Sampling and data gathering a) When was the data for Phase I and for Phase II collected? b) The N for each group (the fieldworkers and field managers) are inconsistent in the abstract and in the methods section of the paper. More specificity is needed, particularly about the field workers. Was there no demographic data collected about these fieldworkers other than living in the community and, usually having about 12 years of formal training? The same holds for the field managers. Can we learn more about their demographics? Note that in the discussion of methods on p. 4, you indicate that 25 fieldworker managers from Phase I were included, instead of 22. c) Study site and research institutions – Out of 34 institutions approached, there was about a 53% response rate or 18 institutions that responded. However, only 12 fieldworkers out of the 18 provided any data. Are these response rates of concern? What kinds of follow-ups were done to augment the size of each group of workers? d) Workshop attendees – 25 managers from 15 of the 18 research centers attended the workshops. But I thought you only had 18 managers who responded positively to be in the phone interviews? e) Points b-d need to be consistent with one another. If not some explanation for the differences should be provided. So you basically used nonprobability sampling. And remember that in your limitations section in the abstract, you mention that there was likely sampling bias from using only respondents from the INDEPTH NETWORK and training department at KWTRP. You go on to say that “This selection is not representative of research institutions that employ fieldworkers in Africa”. So whether the wide range of institutions provide a good overview of how research institutions in Africa manage fieldworkers” seems a questionable conclusion.
---

Analysis: (on p. 6) It seems reasonable, but it is impossible to tell whether the data from the 2.5 day workshop was included in this qualitative analysis. And if so, did you work from tape recordings and transcripts, or something else? 2,5 days of groups transcripts seems quite a lot to manage. Perhaps there were certain portions of the workshop that were analyzed, but your reader doesn't know this.

Results: There needs to be justification of merging the data from the phone interviews from the fieldworkers and the managers with the workshop data. The field workers were not included in the workshop. Perceptions of field workers and managers may have not converged in some areas. Workers and managers certainly represent different points of view and different levels of power. Data from group discussions and individual interviews may come up with different themes. Data from workers and managers should be separated and then compared.

For a qualitative analysis, there are very few quotations (excerpts) provided. There were a total of 5 field worker utterances presented, but representing only 5 different fieldworkers.

More data needs to be reported from the fieldworker interviews if the article is to represent both perspectives. What was common, what obtained little endorsement?

No qualitative excerpts are provided for the workshop or field manager interviews. The points made are interesting, but they are not tied to any specific data. So it is hard for the reader to determine whether there really was the level of consensus on the major points using the actual words of the respondents that is proposed. Did the workers and managers agree on the major themes, or did one group focus more on some than on others? Some of these questions should be reflected in a revised findings section.

Additional references on field workers views and experiences from the US (two different studies – one using focus groups and the other using intensive interviews)

(On P. 13, middle para, it is Gala True, not True Gala).

True, G., Alexander, L.B., & Fisher, C.B. (2017). Supporting the role of community members employed as research staff: Perspectives of community research workers working in addition research.

Social Science & Medicine, 2017)-67-75. (Focus group study)

Richman, K.A., Alexander, L.B., & True, G. (Online first 8/12/14) (2015). How do street- levelresearch workers think about the ethics of doing research “on the ground” with marginalized target populations? AJOB Empirical Bioethics 6 ,2, 1-11. (intensive interview study)

Richman, K.A., Alexander, L.B., & True, G. (2012). Proximity, ethical dilemmas and community research workers. AJOB (American Journal of Bioethics) Primary Research. 3(4), 19-19. (intensive interview study)

Finally and importantly, how does the data gathering and conclusions reported on in this manuscript jive with that reported on in Kombe, F(2015). Enhancing quality and integrity in biomedical research in Africa: An international call for greater focus, investment and standardization in capacity strengthening

	for frontline staff. BMC Medical Ethics, 16:77, listed in the reference section of the current manuscript? The 2015 data came from telephone interviews and a workshop in Kenya in July, 2104. It would appear that no fieldworkers were involved in this report. So that may be one difference between the manuscript under review and the 2015 report. Are there others? The major conclusions from both articles seem rather similar. How does the investigation reported on in the manuscript under review extend and/or contradict what was included in the 2015 report? This would be very instructive to include in the manuscript.
--	--

VERSION 1 – AUTHOR RESPONSE

- In the research question I suggest to incorporate more clearly that this is about the experiences of fieldworkers/workshop members	This has now been clarified in the objectives section both in the abstract and the main manuscript
- If possible, I would add something in the discussion in why education is particularly important in sub-Saharan countries	This has been done under section C. Ways of strengthening institutional support “There appears to be a mismatch between the lack of appropriate training to fieldworkers; fieldworkers education level and the complexity of the tasks they are expected to undertake, including but not limited to translating study information to study participants, understanding institutional and international ethical guidelines and applying these when interacting with study participants. Research institutions in Africa should therefore invest in training and building the capacity of fieldworkers, in order to enhance the standards of the work fieldworkers are employed to undertake and provide optimum support for research”
- This is a qualitative study, so outcomes remain subjective, which is OK - I could not find any limitations of the study/reporting checklist - I could not find a clear data analysis section	A limitation (page 18) and data analysis (page 10) section have been added
In the Introduction the authors write that ‘fieldworkers play a critical role’...; I would rewrite this a bit, in ‘are part of the system that promote scientific and ethical standards’ (more than a role).	This has been done: First sentence under the abstract section
- In either the study objective or introduction, it would help the reader if the authors would give a short definition of a fieldworker. I think it is more or less similar as research assistants in European/US projects?	This has briefly been done under the introduction. A more detailed description is available under the background.

- In the results-section again, I feel that 'role' is maybe not the right term in this context. Perhaps, their tasks (such as...) were common is easier to understand.	This was edited under the "Current Institutional structures and systems for fieldworkers" However, the term "roles" was used throughout the interview and the subsequent workshop hence in some sections, replacing it with "task" may not be appropriate.
- I wondered whether the options in an African setting are completely different compared to a US/European setting, and whether this is specific reason to pay more attention to the fieldworker tasks and education than in European countries, as most people often 'promote' as a researcher. Or whether the problem is especially related to the African/financial context. It would be interesting if the authors could reflect a bit more in the discussion section about this aspect also. - In the results/conclusion I wondered whether some of the fieldworkers also indicated to prefer more education on 'content-specific' topics.	We do appreciate this comment and we acknowledge that Africa setting is different from US/Europe, in terms of education and training. We do not have sufficient experience with the USA/European fieldworker to comprehensively relate our finds to those from those setting. However, there seems to be some similarities when it comes to the ethical challenges experienced by FWs. For example Davis AM, Hull SC, Grady C, Wilfond BS, Henderson GE. The invisible hand in clinical research: The study coordinator's critical role in human subject's protection. The Journal of Law, Medicine & Ethics. 2002 Sep;30(3):411-9). L do identify challenges experiences by community research workers in the USA that are similar to those reported in Africa. Lack of training and limited education may of course influence the FWs ability to deal with these challenges but this argument would need to be supported by some empirical evidence. A brief reflection has been added under the "Responding to complex challenges on the ground" sub-section in findings and discussion section. We indicated that 'content-specific' topics were of course important in helping FWs to undertake study specific role and were appreciated. However, the majority of the challenges fieldworkers experience, that ultimately influenced the quality of their study specific data require more generic and cross-cutting soft-skills and training, which usually lacked in content-specific' topics
General: While reading this method-section I wondered how fieldworkers were selected; what are the inclusion criteria of being a 'fieldworker', are there large differences in tasks across fieldworkers? What kind of research projects I can think of. What type of activities those fieldworkers generally do? Although I understand that the main focus of this paper is on training, I'm very	This information is contained under the "study site and selection of research institutions section" which have now been edited as "study site and selection of research institution and interview participants" to address the issue. An extract of this section reads

interested about what kind of people we are talking about.	From the original list of 34 institutions obtained from the KWTP training department and INDEPTH site, we sent an email to the directors with an information sheet about the study and requested them to identify one senior fieldworker and field manager from their institution, who were able to talk to us about the study. A total of eighteen African international research institutions (see table 1) and the MRC-UK and the INDEPTH Network Secretariat in Ghana responded positively and hence participate The following paragraph has been added Study site and selection of research institutions and interview participants to make this more explicit under The decision to invite institutions affiliated to INDEPTH Network was also informed by the fact that the INDEPTH Network encompasses all research centers in sub-Saharan Africa that have Health Demographic Surveillance Systems which in themselves require a significant amount of fieldwork and are also used as platforms for other field studies.
- Page 6, line 1-9; I wondered what type of research centres were included (could be described in text more clearly)?	This has been done. A paragraph has been added on page 8 explaining the characteristics of the institutions. More details about the institutions can be found on table 1
Can the authors describe the data analysis more concrete?	A section on data analysis has been added
General: I would suggest to split findings and discussion in two separate sections, as this will probably make the interpretation and discussion more easily to follow. I also became a bit confused because of all different ways of numbering.	The numbering has been revised and hope this is now easy to follow. The reason we combined the findings with the discussion was to link the discussion to the specific themes. We believe this makes it much easier for readers to link the discussion directly to the study findings
- Can the authors explain why they decided to present the findings of interviews and the workshop together as well? I can imagine, that it might be good also to present the data of the workshop later-on, as this may partly result from the interviews, and this is a completely different type of data collection. Again, splitting those sections may help in interpreting the data more easily.	Findings of the workshop are already published in the BMC medical ethics journal paper (Kombe F. Enhancing quality and integrity in biomedical research in Africa: an international call for greater focus, investment and standardisation in capacity strengthening for frontline staff. BMC medical ethics. 2015 Dec;16(1):77).

	The current paper does not intend to repeat the data already published in the previous workshop paper. However, where applicable, we tried to compare and contrast the findings from these two phases of the study to give our readers the richness of the views captured across the two different approaches (as intended in our study design). Separating the two sections would limit this and make it difficult to talk about the workshop without the danger of repeating what has already been published. It is to be noted that the majority of those who attended the workshop are the same people who were interviewed during the telephone interviews. This is why some of the views are similar, but we do believe the workshop deliberative discussions were invaluable and provided a rich and wider perspective of the issues under study
- Page 13, line 16 and further. Could the authors give some explanations of the type of difficult/traumatic life experiences? This will facilitate reading.	This has been done under the “Creating space for experience sharing and problem-solving” section page 16
- Page 13, line 48, I can imagine that it very much depends on the type of fieldwork as to whether you need those courses such as GCLP. In this section as well, I’m really thinking of what type of work those fieldworkers need to do.	We appreciate this comment, which supports the observation made on page 17 under the section- “Increasing attention on soft skills for fieldworkers”
- Page 13, line 48; can the authors give some local challenges? I can imagine that this for instance also has to do with culture, access to material, et cetera.	These have been described under the section “Responding to complex challenges on the ground” in page 13
I would first start with the notion that this is all about the experiences/opinions of fieldworkers / workshop-members. E.g. state that fieldworkers themselves believe that more support would improve their skills, and accordingly the quality of the data collection.	The study involved both fieldworkers and fieldworkers’ managers. While we appreciate that the focus was in the performance management experiences of fieldworkers, we feel that both the fieldworkers and fieldworkers’ managers views were important. We believe that this paper managed to present all reviews without giving more weight to the views of the fieldworkers’ than those of the fieldworkers’ managers. In order to address this comment, the following paragraph was added under the Ways of strengthening institutional support In this study, we noted a mismatch related to the lack of appropriate training to fieldworkers;

	fieldworkers' limited level of education and the complexity of the tasks they are expected to undertake, including but not limited to translating study information to study participants, understanding institutional and international ethical guidelines and applying these when interacting with study participants. These findings support the need for research institutions in Africa to invest in training and building the capacity of fieldworkers, in order to enhance the standards of the work fieldworkers undertake and provide optimum support for research. We also added a paragraph under the section Seeing fieldwork as a short-term activity While some field managers supported this view, some argued that career development was a matter of personal motivation and determination. They explained how they [managers] had moved through the carrer ladder to become field managers, often starting their careers as junior fieldworkers and gaining extensive fieldwork experience along their career progression to their current positions, with the same limited training and support. Most managers had therefore been part of the system for a long time and were able to share their views both from the perspective of fieldworkers as well as field managers hence extending the spectrum of experiences adduced and provide deeper insights to the issues that were raised
In the second sentence many aspects are mentioned. I would suggest to focus on a few aspects and then elaborate on this a bit more. I further wondered whether the recommendations are feasible in this setting. Could the authors reflect a bit more on this, also?	Regarding the reflections on the recommendations, this is already available under the conclusion session-Page 18-19 "We realize that the results of this study may not be generalizable, as research institutions were selected based on specific context commonalities and association with the INDEPTH network and KWTRP. However, they do offer important insights on fieldworkers and field managers' views and experiences of fieldworkers' performance managing practices for institutions that come from similar settings as those that participated in this study. Although some of the proposed approaches such as soft-skills training, centralized coordination of fieldworkers training, fieldworkers' scheme of service and regular

	discussions to address fieldwork challenges have been ongoing in institutions such as the KWRTP, in Kenya, lack of documented evidence makes it difficult to say how effective and sustainable such approaches are. More research, therefore, needs to be done to explore the feasibility of the proposed strategies and generalizability of these study findings within a wider context in Africa.”
clarify Ns throughout	This has been done by explaining about the total number of institutions approved, and the proportion of these that agreed to take part in the study (page 8, under Phase 1: Telephone Interviews with managers and fieldworkers) The other Ns under methods have been deleted and written (page 2 under methods)
justification for merging fieldworker data with field manager and workshop data - need to be separated in findings - more excerpts from both managers and workers in findings	Although those interviewed were classified in two categories, it is important to note that the same interview guide was used to interview fieldworkers and fieldworkers’ managers. During the analysis, we compared and contrasted the fieldworkers and fieldworkers’ managers views based on specific themes. Through this, we realized that there were issues where they shared the same views and others where they had different views. In this paper, we tried to present these similarities and contrasts together, rather than separately. We believe this will be easier for our readers to follow and minimise repetition.
How this study extends findings of 2015 report in MMJEthics in 2015?	The BMC Medical Ethics in 2015 paper (Kombe F. Enhancing quality and integrity in biomedical research in Africa: an international call for greater focus, investment and standardization in capacity strengthening for frontline staff. BMC medical ethics. 2015 Dec;16(1):77) concerns the deliberative workshop proceedings only. The current paper present data of the whole study, which included the telephone interviews and to a less extend, some insights from the workshop to deepen the richness of the methods and discussions help. It is important to note that, the workshop discussed in details the findings from the first phase with a view to reach consensus and identify strategies to address the challenges identified during the first phase. The workshop was therefore

	strengthened the methods used and further extended the data collected during the telephone interviews. Because of this, there are certain areas where we tried to compare and contrast the findings from these two phases of the study to give our readers the richness of the views captured through the two different approaches. Separating the two sections would limit this and make it difficult to talk about the workshop without the danger of repeating what has already been published. A paragraph under the methods section has been added to make this clarification “Although this paper aims to focuses more on the results from Phase I, it is important to note that the same participants who attended the workshop were also interviewed during the telephone interviews (Phase I). There may therefore be some overlaps in certain sections of these papers. It is also important to note that in Phase II (the workshop} issues raised in phase 1 were re-discussed in more detail with a view to build consensus and identify common recommendations and strategies of addressing the reported challenges. As such, there are certain sections that inevitably refer to views shared during the workshop. It is however, not the intention of the authors to publish the phase II results of this study, (12) that have already been published.
When was the data for Phase I and for Phase II collected?	Phase II was conducted early 2014 And Phase II was conducted in July 2014
b) The N for each group (the fieldworkers and field managers) are inconsistent in the abstract and in the methods section of the paper. More specificity is needed, particularly about the fieldworkers. Was there no demographic data collected about these fieldworkers other than living in the community and, usually having about 12 years of formal training? The same holds for the field managers. Can we learn more about their demographics?	The N typo has been corrected. We appreciate this comment on demographic data. Unfortunately, we deliberately did not collect individual demographic data to avoid instilling fear of being linked to the data and victimization, if participants shared views that had the potential to be viewed negatively by institutional managements (see interview guide for information collected). This was particularly important given the participants were referred to us by the centre directors and partly the reason why names of the centres that participated in the study are anonymized.
Note that in the discussion of methods on p. 4, you indicate that 25 fieldworker managers from Phase I were included, instead of 22.	Thank you for the comment. The numbers has now been reviewed.

c) Study site and research institutions – Out of 34 institutions approached, there was about a 53% response rate or 18 institutions that responded. However, only 12 fieldworkers out of the 18 provided any data. Are these response rates of concern? What kinds of follow-ups were done to augment the size of each group of workers?	The study did not aim to interview a representative sample of either of the two groups (fieldworkers and fieldworkers’ managers), rather to explore in depth, the views and experiences of fieldworkers and fieldworkers’ managers. The aim was to interview at least ONE fieldworker and ONE fieldworkers’ manager from each of the research institutions. We therefore believe that the response rate was very good and helped us to get views and experiences from fieldworkers and fieldworkers’ managers from a range of institutions with different characteristics (see table 1) We also feel that the value of this study is not that every research centre in Africa can identify with every issue raised here, this would be difficult due to the variety of research centres in the continent, rather, we hope that by raising the issues highlighted here that even those centre not involved in the study will be provoked to examine their fieldworkers support system and see if any issues raised here apply to their setting
d) Workshop attendees – 25 managers from 15 of the 18 research centers attended the workshops. But I thought you only had 18 managers who responded positively to be in the phone interviews?	Yes 25 managers from 15 of the 18 research centers attended the workshops. In addition to the interviewees who had been invited, there were additional managers from the KWTRP- where the workshop was held, including the Head of training, three senior social scientists with long term experience in managing fieldworkers (also Co-PIs of the project), the Head of community engagement and one fieldworker supervisor. To make this more explicit-Part of the methods section in page 5 has been edited to read in part as “In Phase two, we held a two-day workshop involving 25 fieldworker managers In Phase two, we held a two-day workshop involving 25 delegates including the 18 (one per institution) from the institutions that were involved in phase I; and the director of training, three senior social scientists with long term experience in managing fieldworkers (also Co-PIs of the project), the Head of community engagement and two senior fieldworkers’ supervisor from the KWTRP- Total 25 participants. A training expert and an independent consultant with experience in

	group moderation and consensus building techniques facilitated the workshop.
e) Points b-d need to be consistent with one another. If not some explanation for the differences should be provided.	This has been done as above
So, you basically used non-probability sampling. And remember that in your limitations section in the abstract, you mention that there was likely sampling bias from using only respondents from the INDEPTH NETWORK and training department at KWTRP. You go on to say that "This selection is not representative of research institutions that employ fieldworkers in Africa". So, whether the wide range of institutions provide a good overview of how research institutions in Africa manage fieldworkers" seems a questionable conclusion	Indeed, we do acknowledge this as a limitation and in our conclusion, we caution our readers that the findings from this study may not be generalizable given the stated limitation. We however underscored the fact that the institutions were quite diverse, with 7 Research Institution coming from West Africa, 4 Research Institution from Southern Africa and 7 Research Institution from East Africa, which gives us confidence that the findings from this study can offer important insights for institutions that come from similar settings as those that participated in this study (see conclusion section on page 18) .
Analysis: (on p. 6) It seems reasonable, but it is impossible to tell whether the data from the 2.5 day workshop was included in this qualitative analysis. And if so, did you work from tape recordings and transcripts, or something else? 2,5 days of groups transcripts seems quite a lot to manage. Perhaps there were certain portions of the workshop that were analyzed, but your reader doesn't know this.	The detailed description of how the workshop data was analyzed was already published in the BMC Medical ethics paper. The data analysis section on page 10 of this paper explains explicitly how the analysis was carried out.
Results: There needs to be justification of merging the data from the phone interviews from the fieldworkers and the managers with the workshop data. The field workers were not included in the workshop. Perceptions of field workers and managers may have not converged in some areas. Workers and managers certainly represent different points of view and different levels of power. Data from group discussions and individual interviews may come up with different themes. Data from workers and managers should be separated and then compared. For a qualitative analysis, there are very few quotations (excerpts) provided. There were a total of 5 field worker utterances presented, but representing only 5 different fieldworkers. More data needs to be reported from the fieldworker interviews if the article is to represent both perspectives. What was common, what obtained little endorsement? No qualitative excerpts are provided for the workshop or field manager interviews. The points made are	Although those interviewed were classified in two categories, it is important to note that the same interview guide was used to interview fieldworkers and fieldworkers' managers. It was therefore logical to analyze and present these data together. During the analysis, we compared and contrasted the fieldworkers and fieldworkers' managers views based on specific themes. Through this, we realized that there were issues that they shared the same views and others that they had different views. In this paper, we tried to present these similarities and contracts together, rather than separately. We believe this will be much easier for our readers to follow than it would have been if we separated the views of fieldworkers from those of the fieldworkers' managers.

interesting, but they are not tied to any specific data. So, it is hard for the reader to determine whether there really was the level of consensus on the major points using the actual words of the respondents that is proposed. Did the workers and managers agree on the major themes, or did one group focus more on some than on others? Some of these questions should be reflected in a revised findings section.	
Additional references on field workers views and experiences from the US (two different studies – one using focus groups and the other using intensive interviews) (On P. 13, middle para, it is Gala True, not True Gala). True, G., Alexander, L.B., & Fisher, C.B. (2017). Supporting the role of community members employed as research staff: Perspectives of community research workers working in addition research. Social Science & Medicine, 2017)-67-75. (Focus group study), Richman, K.A., Alexander, L.B., & True, G. (Online first 8/12/14) (2015). How do street level research workers think about the ethics of doing research “on the ground” with marginalized target populations? AJOB Empirical Bioethics 6 ,2, 1-11. (intensive interview study) Richman, K.A., Alexander, L.B., & True, G. (2012). Proximity, ethical dilemmas and community research workers. AJOB (American Journal of Bioethics) Primary Research. 3(4), 19-19. (intensive interview study)	This has been well appreciated and added
Finally, and importantly, how does the data gathering and conclusions reported on in this manuscript jive with that reported on in Kombe, F(2015). Enhancing quality and integrity in biomedical research in Africa: An international call for greater focus, investment and standardization in capacity strengthening for frontline staff. BMC Medical Ethics, 16:77, listed in the reference section of the current manuscript? The 2015 data came from telephone interviews and a workshop in Kenya in July, 2104. It would appear that no fieldworkers were involved in this report. So that may be one difference between the manuscript under review and the 2015 report. Are there others? The major conclusions from both	The current paper does not intend to repeat or extend the data already published in the previous workshop paper but present data of the whole study, which included the telephone interviews as well as the workshop. In certain areas of this new paper; and where applicable, we tried to compare and contrast the findings from these two phases of the study to give our readers the richness of the views captured through the two different approaches. Separating the two sections would limit this and make it difficult to talk about the workshop without the danger of repeating what has already been published.

articles seem rather similar. How does the investigation reported on in the manuscript under review extend and/or contradict what was included in the 2015 report? This would be very instructive to include in the manuscript.	
--	--

VERSION 2 – REVIEW

REVIEWER	Leslie B. Alexander Bryn Mawr College
REVIEW RETURNED	27-Jun-2019

GENERAL COMMENTS	The authors made many of the requested changes and were responsive to questions raised by the reviewers..The paper is much more straightforward. I still have concerns about some of the limitations of the study which I address further in the attached file Overall, the authors provided thoughtful responses to the questions raised in the first review, resulting in a much more straightforward and nuanced presentation. The flow and organization of the article is greatly improved. There remain a few areas of concern: Under the limitations section: 1) (p. 5, l. 8-10. Saying that all interviews were done by phone and then concluding that “We do believe that this did not affect the quality of the data collected” is a weak argument. On what bases was such a conclusion reached? 2) This is an assertion only. Is this point really needed? Why not just acknowledge that fieldworkers’ responses could have been biased – You did what you could to try and prevent this, but the field workers and staff were selected by the Center Directors. The power differentials between these two groups of workers are large and one would expect that their perspectives might differ in some ways because of their differences in location in the organization. Likewise they were all selected by the agency directors. There was also no attempt to look at dyads of managers and field workers by agency. There was certainly a lot of variation in these agencies which could have influenced the discussion of both managers and field workers. (See page 16, lines , l. 31-48) 3) P. 5, l.37-47 The generalizability issue is raised here, with an assertion that agencies that were not included in the study, if they had similar characteristics to those in the sample, could result in generalizable results beyond the sample. What sample characteristics are key here? A more measured conclusion, as presented in the conclusions section of the paper (pp. 24 ff) seems more justified. This is an exploratory study, with a convenience sample and any hint that the results are generalizable in any distributional sense is not justified. That doesn’t mean that the conceptual findings couldn’t guide future research. Why not acknowledge that the study is exploratory.
---

	As earlier, my concern remains about mixing the data of the field workers and staff in the findings. Since the recommendations were guided by the interview data, which is what is being focused on here, why not say that directly in the limitations? Why wasn't a workshop performed with the field workers as well? Since that group was half the size of the manager group, couldn't a workshop devoted to the data generated in their interviews have occurred but been shorter? Under Methods, you could reiterate that no field workers were included in the 2.5-day workshop. You say it in the beginning, but then there is some slippage in this regard in other parts of the results. There are significant power differentials between the workers and the managers p. 13 – line 3. Weren't there many more questions in the field manager interviews than in the field worker interviews? If that is so, it seems odd that all interviews lasted for about 90 minutes. Field managers had additional questions to respond to according to the interview guide presented in the first review. See p. 15, lines 52-56. It seems contradictory. p. 21 –l. 7-12 – Field workers were not included in the workshop. p. 23, l.32 – It is True (2011), not Galas.; same issue on p. 24, l.32 – Should be True, not Gala et al.rue p. 24 – l. 8 There should be a citation for the previous commentary. In the justification for the article under review being different from the 2015 article which reported the findings of the workshop, it is referred to in the authors' response to the review critique as "proceedings". That doesn't seem quite right. A commentary seems better, though you could also argue that it was exploratory research. It seemed systematic.
--	---

VERSION 2 – AUTHOR RESPONSE

Reviewer 2 comments	
The authors made many of the requested changes and were responsive to questions raised by the reviewers. The paper is much more straightforward. I still have concerns about some of the limitations of the study which I address further in the attached file	We appreciate the reviewer's comment. The comments on the limitation has been addressed as below
Under the limitations section: 1) (p. 5, l. 8-10. Saying that all interviews were done by phone and then concluding that "We do believe that this did not affect the quality of the data collected" is a weak argument. On what bases was such a conclusion reached?	 This has been edited to "All interviews were done by phone. As such, important interview dynamics may have been missed. Face to face interviews would have been able to pick important non-verbal cues, which was not possible through telephone interviews" (page 3)
This is an assertion only. Is this point really needed? Why not just acknowledge that fieldworkers' responses could have been biased – You did what you could to try and prevent this, but the field workers and staff were selected by the Center Directors. The power differentials between these two groups of workers are large and one	 This was edited to "interviewees were nominated by Centre directors, who were the primary contact people. Although individual consent was obtained from those nominated by the directors prior to the interview, it is possible that interviewees could have reported what they believed would be acceptable to their directors. During the interviews, we assured our

would expect that their perspectives might differ in some ways because of their differences in location in the organization. Likewise they were all selected by the agency directors. There was also no attempt to look at dyads of managers and field workers by agency. There was certainly a lot of variation in these agencies which could have influenced the discussion of both managers and field workers. (See page 16, lines , l. 31-48)	participants about the confidentiality of the data they would share, which we believe enabled them to share information sincerely and transparently. However, there is a potential that the power differentials between the participants (field managers and fieldworkers), and the centre directors could have influenced their views and perceptions of the discussion held. Despite fieldworkers being hierarchically lower, compared to field managers, with different powers and authority within the system, we could not identify any strong divergence in the views they shared. ” Page 3
P. 5, l.37-47 The generalizability issue is raised here, with an assertion that agencies that were not included in the study, if they had similar characteristics to those in the sample, could result in] generalizable results beyond the sample. What sample characteristics are key here? A more measured conclusion, as presented in the conclusions section of the paper (pp. 24 ff) seems more justified. This is an exploratory study, with a convenience sample and any hint that the results are generalizable in any distributional sense is not justified. That doesn't mean that the conceptual findings couldn't guide future research. Why not acknowledge that the study is exploratory.	This was edited in part, to “It is possible that other institutions outside these networks had different experiences from those that participated in this study. We appreciate that this is an exploratory study. We therefore believe the institutions involved shared a wide range of experiences and that findings from this study offer important insights for further research, especially for international health research institutions that employ fieldworkers” (page 4)
As earlier, my concern remains about mixing the data of the field workers and staff in the findings. Since the recommendations were guided by the interview data, which is what is being focused on here, why not say that directly in the limitations? Why wasn't a workshop performed with the field workers as well? Since that group was half the size of the manager group, couldn't a workshop devoted to the data generated in their interviews have occurred but been shorter? Under Methods, you could reiterate that no field workers were included in the 2.5-day workshop. You say it in the beginning, but then there is some slippage in this regard in other parts of the	We appreciate this comment and we would have loved to have had a workshop with the fieldworkers as well, but due to funding constrains, we were only able to hold one workshop. We could not combine managers and the fieldworkers, due to the power differential issues raised above. Further work is however being done with fieldworkers, following the findings from the workshop, where fieldworkers are more closely involved. This has been edited in part to “Fieldworkers were not involved in the second phase” (Page 6)

results. There are significant power differentials between the workers and the managers	
p. 13 – line 3. Weren't there many more questions in the field manager interviews than in the field worker interviews? If that is so, it seems odd that all interviews lasted for about 90 minutes. Field managers had additional questions to respond to according to the interview guide presented in the first review. See p. 15, lines 52-56. It seems contradictory.	On page 10, we stated that "Each telephone interview lasted for approximately one and a half hours". 90 minutes is therefore an approximation, but it's true some interviews took shorter or longer than 90 minutes but we did not analyse this by the two groups.
p. 21 – l. 7-12 – Field workers were not included in the workshop.	This has been edited to "During workshop discussions about support mechanisms to help fieldworkers address these emotional challenges, field managers were not able to identify existing strategies; as was also reported by fieldworkers during the telephone interviews (Page 16)
p. 23, l.32 – It is True (2011), not Galas.; same issue on p. 24, l.32 – Should be True, not Gala et al.rue	This has been edited Page 19
p. 24 – l. 8 There should be a citation for the previous commentary. In the justification for the article under review being different from the 2015 article which reported the findings of the workshop, it is referred to in the authors' response to the review critique as "proceedings". That doesn't seem quite right. A commentary seems better, though you could also argue that it was exploratory research. It seemed systematic.	This has been edited to include the reference page 19 This comment is noted.

VERSION 3 – REVIEW

REVIEWER	Leslie B. Alexander Bryn Mawr College
REVIEW RETURNED	16-Sep-2019

GENERAL COMMENTS	2nd Revision -2018-028453 – Enhancing Fieldworkers' Performance Support..... Some final issues: 1)p. 3, l.33 - Under Phase II - including 18 field managers from the institutions that were involved 2)p. 3. L. 37 – The first reference to the BMC Medical Ethics (2015) article should go here. (You could say that an earlier report from Phase II appeared in..... There is no way to get around that there was some analysis in this earlier article of this consultative workshop 3)p. 4 – l 35 –this may provide some support for..... You mention the potential for power differentials on p. 4, ol.59, which challenges whether the two perspectives are fungible with each other. You
--

	can note that there wasn't a strong divergence but you have also talked about the potential for social desirability bias as well.. 4)p 9, l. 3 – ; thus the reasons for nonresponse are unknown. 5) p. 14 , l. 48. While there were some similar questions, the field managers' protocol was considerably longer, allowing for more depth of response.(It seems plausible that some of the field manager interviews could have been1.5 hours. It seems unlikely that most of the field workers' interviews were that long. 6) I am assuming that the Appendix with the interview schedules will be included in the final draft. You provided more detail about the length of the interviews in the original draft. I would suggest going back and including these data about the period of time the interviews took place and the mean, median and SD for those data. It seems likely that the distribution was skewed. 7) The initials after the few direct quotes included are not clear. To what do the two sets of initials refer? 8) p. 24 –l.8. This is the first mention of using skype. What percentage of the interviews were done by skype? That medium approximates a face to face interview, if the technology on both ends is equally good. If you can document how many field workers and managers had skype interviews, then keep it. Otherwise, leave it out. The justification of folding the two types of data together remains a concern for this reader. At least the argument has been tempered a bit and acknowledgement made about power differentials and the effects of potential social desirability.. The generalizability of findings has been improved.
--	---

VERSION 3 – AUTHOR RESPONSE

1)p. 3, l.33 - Under Phase II - including 18 field managers from the institutions that were involved	This has been edited on page 3
2)p. 3. L. 37 – The first reference to the BMC Medical Ethics (2015) article should go here. (You could say that an earlier report from Phase II appeared in..... There is no way to get around that there was some analysis in this earlier article of this consultative workshop	In text referencing has been done. However, since this is under the abstract section, we could not insert the actual reference per se.
3)p. 4 – l 35 –this may provide some support for..... You mention the potential for power differentials on p. 4, ol.59, which challenges whether the two perspectives are fungible with each other. You can note that there wasn't a strong divergence but you have also talked about the potential for social desirability bias as well..	This has been edited on page 4
4)p 9, l. 3 – ; thus the reasons for nonresponse are unknown.	This has been edited on page 9
5) p. 14 , l. 48. While there were some similar questions, the field managers' protocol was considerably longer, allowing for more depth of response. (It seems plausible that some of the field manager interviews could have been1.5 hours. It	This has been edited on page 9

seems unlikely that most of the field workers' interviews were that long.	
6) I am assuming that the Appendix with the interview schedules will be included in the final draft. You provided more detail about the length of the interviews in the original draft. I would suggest going back and including these data about the period of time the interviews took place and the mean, median and SD for those data. It seems likely that the distribution was skewed.	The appendix is included. However, we did not conduct any quantitative analysis hence unable to provide the mean, median and SD of the time the interviews took, as recommended by the reviewer
7) The initials after the few direct quotes included are not clear. To what do the two sets of initials refer?	A footnote to explain the initials has been added on 14
8) p. 24 –1.8. This is the first mention of using skype. What percentage of the interviews were done by skype? That medium approximates a face to face interview, if the technology on both ends is equally good. If you can document how many field workers and managers had skype interviews, then keep it. Otherwise, leave it out.	VOIP stated on page 7 under data collection includes Skype all but we agree this was not explicitly stated under this section. As advised by the reviewer, we have deleted skype as it was indeed appearing here for the first time and that we feel its removal does not make any difference as we consider this very similar to telephony calls
The justification of folding the two types of data together remains a concern for this reader. At least the argument has been tempered a bit and acknowledgement made about power differentials and the effects of potential social desirability. The generalizability of findings has been improved.	We did our best in giving a reasonable justification for folding the two types of data together. We thank the reviewers for their constructive feedback and comments and believe this approach will allow our readers gain a good understanding of the study and the approaches used.